# Expression of Stress-Mediating Genes is Increased in Term Placentas of Women with Chronic Self-Perceived Anxiety and Depression

**DOI:** 10.3390/genes11080869

**Published:** 2020-07-31

**Authors:** Cristina A. Martinez, Ina Marteinsdottir, Ann Josefsson, Gunilla Sydsjö, Elvar Theodorsson, Heriberto Rodriguez-Martinez

**Affiliations:** 1Department of Biomedical & Clinical Sciences (BKV), BKH/Obstetrics & Gynaecology, Faculty of Medicine and Health Sciences, Linköping University, SE-58185 Linköping, Sweden; ann.josefsson@regionostergotland.se (A.J.); Gunilla.Sydsjo@regionostergotland.se (G.S.); heriberto.rodriguez-martinez@liu.se (H.R.-M.); 2Department of Medicine and Optometry, Faculty of Health and Life Sciences, Linnaeus University, Hus Vita, 43157 Kalmar, Sweden; ina.marteinsdottir@liu.se; 3Division of Clinical Chemistry, Department of Biomedical and Clinical Sciences, Faculty of Medicine and Health Sciences, Linköping University, SE-58185 Linköping, Sweden; gudjon.elvar.theodorsson@liu.se

**Keywords:** antenatal stress, hair cortisol, term-placentae, RT-qPCR, human

## Abstract

Anxiety, chronical stress, and depression during pregnancy are considered to affect the offspring, presumably through placental dysregulation. We have studied the term placentae of pregnancies clinically monitored with the Beck’s Anxiety Inventory (BAI) and Edinburgh Postnatal Depression Scale (EPDS). A cutoff threshold for BAI/EPDS of 10 classed patients into an Index group (>10, *n* = 23) and a Control group (<10, *n* = 23). Cortisol concentrations in hair (HCC) were periodically monitored throughout pregnancy and delivery. Expression differences of main glucocorticoid pathway genes, i.e., corticotropin-releasing hormone (CRH), 11β-hydroxysteroid dehydrogenase (HSD11B2), glucocorticoid receptor (NR3C1), as well as other key stress biomarkers (Arginine Vasopressin, AVP and O-GlcNAc transferase, OGT) were explored in medial placentae using real-time qPCR and Western blotting. Moreover, gene expression changes were considered for their association with HCC, offspring, gender, and birthweight. A significant dysregulation of gene expression for CRH, AVP, and HSD11B2 genes was seen in the Index group, compared to controls, while OGT and NR3C1 expression remained similar between groups. Placental gene expression of the stress-modulating enzyme 11β-hydroxysteroid dehydrogenase (HSD11B2) was related to both hair cortisol levels (Rho = 0.54; *p* < 0.01) and the sex of the newborn in pregnancies perceived as stressful (Index, *p* < 0.05). Gene expression of CRH correlated with both AVP (Rho = 0.79; *p* < 0.001) and HSD11B2 (Rho = 0.45; *p* < 0.03), and also between AVP with both HSD11B2 (Rho = 0.6; *p* < 0.005) and NR3C1 (Rho = 0.56; *p* < 0.03) in the Control group but not in the Index group; suggesting a possible loss of interaction in the mechanisms of action of these genes under stress circumstances during pregnancy.

## 1. Introduction

Antenatal maternal stress such as anxiety and depression have been widely associated with short- and long-term negative impact on the neurobiological and physiological functioning of the offspring [1,2,3]. Maternal distress during pregnancy and puerperium, which in Sweden has a 15% incidence [4,5,6], is apparently linked to newborns being at high risk of reduced birthweight, smaller head circumference, and adverse neurodevelopmental outcomes; including increased hypothalamic–pituitary–adrenal axis (HPA) sensitivity, anxiety, depressive-like behaviors, attention deficit, hyperactivity, poor cognitive or emotional developmental disorder, among others [7,8,9]. The growing morbidity among offspring born from stressed pregnancies has been related to both newborn sex and the timing of action of stressors during pregnancy. Gerardin et al. (2011), reported that sons of mothers suffering from depression during pregnancy showed deficits in the regulation of motor skills and of behavior, whereas changes of the latter were not detected in daughters [10]. Additionally, the odds for schizophrenia was higher among children born from mothers stressed at mid-pregnancy than from mothers were exposed to stressors during late pregnancy [11,12,13]. Stress factors during pregnancy lead to hypercortisolemia, poor reactivity, and major circadian rhythm fluctuations of cortisol [14]. Maternal cortisol passes physiologically into the fetus yet being mostly inactivated by the placenta thus avoiding deleterious effects [15,16]. However, from week 20 of gestation, cortisol presence induces placental production of CRH which increases fetal cortisol production [17]. Of interest, it is during this period when the maternal HPA-system becomes increasingly refractory to stress, so the fetus may not be as prone to fluctuations of maternal cortisol. We have recently monitored maternal cortisol during pregnancy, in primiparous and multiparous women in hair (HCC), which better reflects its overall long-time biological activity. Cortisol levels were determined as ascending during pregnancy, to dip more strongly in multiparous 3 months prior to partum compared to primiparous, thus suggesting a quicker suppression of the hypothalamic CRH production by placental CRH in multiparous women (Marteinsdottir et al., manuscript). Secretion of CRH is regulatory of several physiological processes in the fetus, from nervous system shaping to influencing delivery time and further, post-natal development [18]. However, under situations of maternal stress, placental CRH increases significantly, and can be considered a suitable biomarker for fetal distress [19]. Nonetheless, the mechanisms underlying the manifestation of these maternal stressors in the placenta need to be further investigated.

Placental mechanisms underlying stress-induced elevations in maternal glucocorticoids as a key mechanism of stress transmission to the fetus [20] remain yet unclear, in the light of other maternal systems mediating stress responses beyond the HPA-system, i.e., the arginine vasopressin/oxytocin (AVP) or the autonomic nervous (ANS) systems. Maternal stress influencing placental function via glucocorticoid (GC) act via specific receptors (NR3C1) [21,22] and the modulatory enzyme 11b-hydroxysteroid dehydrogenase type 2 (11-β-HSD-2) [15,16]; altering expression of HPA-related genes which rule the stress-modulating enzyme O-LcNAc transferase (OGT) [23].

Here, we hypothesize that perceived maternal stress during pregnancy is capable of triggering stress-related molecular signaling in term placenta which could potentially affect fetal developmental outcomes, related to offspring gender/weight. To test this hypothesis, we determined the term-placental expression of main glucocorticoid pathway genes, i.e., corticotropin-releasing hormone (*CRH*), 11β-hydroxysteroid dehydrogenase (*11β-HSD2*), glucocorticoid receptor (*NR3C1*), as well as other key stress biomarkers (Arginine Vasopressin, *AVP* and O-GlcNAc transferase, *OGT*) and their association with maternal HCC levels, offspring gender, and birth weight.

## 2. Materials and Methods

### 2.1. Ethical Considerations

The integrity of the patients has been granted by Ethical permits warranting full information prior to consent and full anonymity. Data was treated at group levels. No individual is to be identifiable in the publication. This is an established procedure in Swedish clinical investigations and it is fully described in the Ethical Perspectives in Neurology section (EPNs) permissions already obtained. All examinations and tests are harmless and have been used in several previous clinical studies. All data was treated coded and anonymously. The epidemiological surveys are already approved by the Human Research Ethics Committee Linköping (03-556, 07-M66 08–08-M 233-8, 2017/513-31). The study was approved by the Regional Ethical Review Board in Linköping (nr 2011/499-31 and 2013/355-32).

### 2.2. Experimental Design

A total of 390 pregnant women attending an antenatal care clinic in southeast Sweden were included in the study. The women completed anxiety and depression inventories and underwent hair cortisol collection on week 24–25, during childbirth and postpartum. Self-perceived symptoms of anxiety were assessed with the Beck’s Anxiety Inventory (BAI) [24] and symptoms of depression were assessed with the Edinburgh Postnatal Depression Scale (EPDS) [25]. Both inventories are well known, easy to use, validated in Sweden [26], and often used in research settings and as screening in clinical settings [27]. As a measure of symptoms of depression and anxiety a cut off score of 10 was used for both the EPDS and the BAI. By selecting the above threshold the sensitivity for the detection of major depression was nearly 100% and the specificity 82% [28]. Women displaying pregnancy complications, including pre-eclampsia and/or preterm birth were excluded.

A total of 23 women scored > 10 at both EPDS/BAI—indicating symptoms of depression and anxiety are here referred to as index women. A total of 23 controls who scored < 10 on both EPDS/BAI were randomly selected from the entire study population (*n* = 390). After childbirth, the placentae were immediately collected and stored at −80 °C until further analysis. Data on obstetric and neonatal outcomes were collected from standardized medical records. The demographic data of the patients included in this study is shown in Table 1.

### 2.3. Hair Cortisol Concentrations (HCC)

Hair bundles (3 mm thick × 3 cm length) were cut at scalp level from the back of the head (vertex area). The bundles were cut in one-cm segments (5–6 mg), assuming each segment reflected one-month time [29]; placed in canisters holding 2 mL Eppendorf tubes and frozen in liquid nitrogen (LN_2_) until analyzed. A competitive radioimmunoassay (RIA) was used to quantify (pg/mg) cortisol concentrations in methanol-extracted pulverized hair as described by Morelius et al. (2004) [30]. In brief, the weighed (Sartorius MC 210p microscale, Qiagen, Hilden, Germany) hair samples were pulverized (Retsch Tissue Lyser II, Haan, Germany) at 23 Hz for 2 min before adding 1 mL of methanol (Chromasolv, Sigma-Aldrich, Darmstadt, Germany) to extract the cortisol at RT for 10 h. Following centrifugation (Microcentrifuge Thermo Scientific, Waltham, MA, USA, 13,000× *g*, 4 °C for 1 min), 800 μL of the supernatant was lyophilized (SpeedVac Plus SC210A, Savant, Coral Springs, FL, USA) for at least 3 h. The samples were dissolved in radioimmunoassay buffer and RIA-analyzed using an antiserum cross-reacting < 1% with endogenous steroids, <40% with prednisolone and 21-deoxycortisol, and 137% with 5α-dihydroxycortisol [31].

### 2.4. Collection and Preparation of Placenta Samples

The placentas were, immediately after expulsion, placed on ice and a 1.5-cm square of the placental disk dissected, approximately 5-cm apart from the insertion of the umbilical cord. This approx. 2.5-cm thick villous parenchyma was then punched into centripetal >1 g samples of the fetal (including the chorionic plate), the middle, and the maternal (including the thin basal plate) areas, snap-frozen and packed for final storage at −80 °C.

### 2.5. RNA Extraction

Total RNA was isolated from pools of four different segments of placenta samples retrieved from the fetal side using Trizol reagent (Invitrogen, Carlsbad, CA, USA) according to the manufacturer’s instructions. Briefly, tissue samples were mechanically disrupted in 1 mL Trizol reagent using a TissueLyser II (Qiagen, Hilden, Germany). The homogenized tissues were centrifuged at 12,000× *g* at 4 °C for 10 min. Then, supernatants were incubated with bromochloropropane (100 µL/mL homogenized) for 5 min at room temperature. Samples were then centrifuged at 12,000× *g* at 4 °C for 15 min. The aqueous phases obtained were mixed with isopropanol and RNA precipitation solution (1.2 M NaCl and 0.8 M Na_2_C_6_H_6_O_7_) and incubated at room temperature for 10 min. Then, samples were centrifuged at 12,000× *g* at 4 °C for 10 min. After discarding the supernatant, 1 mL of 75% ethanol was added to the pellet fraction and centrifuged at 7500× *g* at 4 °C for 5 min. The RNA pellets obtained were air-dried for 30 min and mixed with 30 µL of RNase free water. The obtained total RNA was quantified with a NanoDrop ND-1000 (Thermo Fisher Scientific, Fremont, CA, USA) and quality with an Agilent 2100 Bioanalyzer (Agilent Technologies, Santa Clara, CA, USA), yielding 8–10 RNA-integrity number (RIN) values.

### 2.6. Protein Extraction

Proteins from placental samples were isolated as previously described [32]. Briefly, 200 μL of RIPA buffer (Sigma-Aldrich, Darmstadt, Germany) mixed with 2 μL of protein cocktail inhibitor (Thermofisher Scientific, Fremont, CA, USA) was added to each sample prior to sonication (Amplitude 50 W, 140). Then, samples were incubated at 4 °C for 60 min in rotation and later centrifugated at 13,000× *g* at 4 °C for 10 min. After centrifugation, the supernatants were collected and proteins were quantified using a DC Protein assay kit (Bio Rad, Hercules, CA, USA), following manufacturer’s instructions. Protein suspensions were denatured by heating at 70 °C for 10 min and kept at −20 °C until analyses.

### 2.7. Relative Quantitative Reverse Transcriptase Polymerase Chain-Reaction (qRT-PCR)

Total RNA was transcribed into cDNA using with 25 mM dNTPs Mix, RT random primers, 20 U of RNase inhibitor and MultiScribe Reverse Transcriptase (High Capacity cDNA Reverse Transcription Kit, Applied Biosystems, Foster City, CA, USA). The qRT-PCR was performed in 10-μL reactions with 5 µL of PowerUp™ SYBR™ Green Master Mix (Applied Biosystems™, Foster City, CA, USA), 50 nM for each set of primers, 2 µL of synthetized cDNA, and water to a final volume of 10 µL. All reactions were carried out using the Real-Time PCR Detection System (CFX96™; Bio-Rad Laboratories, Inc; Richmond, CA, USA). The thermal cycling profile was 50 °C for 2 min, 95 °C for 10 min, 40 cycles at 95 °C for 15 s, and 60 °C for 1 min. Melt curve analysis was carried out to evaluate the specificity of each PCR reaction by detection of one single peak on the dissociation curve profile. The gene relative expression levels were quantified using the (2^−ΔΔct^) [33] method and *GAPDH* as a reference gene for cDNA normalization. Primer sequences are detailed in Table 2.

### 2.8. Western Blotting (WB)

To prepare Western blots, 10 μL (5 μg) of each protein suspension was loaded into 4–20% Mini-PROTEAN^®^ TGX™ Precast Protein Gels (BioRad, Richmond, CA, USA) and transferred to polyvinyldifluoride (PVDF) membranes (BioRad, Richmond, CA, USA). Membranes were then incubated for 1 h in Odyssey Blocking solution (LI-COR Biosciences, Lincoln, NE, USA) and washed 3 × 10 min in washing buffer (Tris-phosphate-buffered saline) containing 0.1% Tween-20 (Sigma-Aldrich, Madrid, Spain). Then, membranes were incubated at 4 °C overnight with the primary antibodies (anti-CRH; rabbit polyclonal antibody (LSBio-B11889, LSBio, Seattle, WA, USA), anti-AVP polyclonal antibody (MBS9205129, MyBiosources, San Diego, CA, USA), and anti-HSD11B2 polyclonal antibody (ab80317, abcam, Cambridge, UK) at 1:1000, 1:500, and 1:1000 dilution rate, respectively. The day after, the membranes were washed 3 × 10 min and incubated for 1 h with the reference anti-GAPDH rabbit polyclonal primary antibody (ab181602, abcam, Cambridge, UK) at 1:10,000 dilution rate, washed again 3 × 10 min and finally incubated with a secondary antibody (goat anti-rabbit IRDye 800 CW, LI-COR Biosciences, Lincoln, NE, USA) at a 1:20,000 dilution on blocking buffer. After extensive washing, the membranes were scanned (Odyssey CLx (LI-COR Biosciences, Lincoln, NE, USA), to obtain blot-images using the Image Studio 4.0 software (LI-COR Biosciences, Lincoln, NE, USA). Raw data comparisons were made only within each blot.

### 2.9. Statistical Analysis

Statistical analyses were conducted using SPSS statistical software (version 24.0; SPSS Inc., Chicago, IL, USA). Gene expression data were analyzed for normality of residuals using the Kolgomorov–Smirnov test. Since data were not normally distributed, the Mann–Whitney U-test was used to analyze the data. Protein expression data were normalized with an endogenous control and statistical significance was determined using Student’s *t*-test. Associations of maternal cortisol levels with placental gene expression and associations among gene expression patterns between groups were analyzed using Spearman’s rho. Gene expression values were represented as log FC (2^−ΔΔct^) (violin plots) or FC (2^−ΔΔct^) (graphs). Differences were considered significant at *p* < 0.05.

## 3. Results

### 3.1. Prenatal Stress Influenced Gene Expression of Term-Placentas

After qPCR analyses, the data obtained was visualized by a principal component analysis (PCA) plot, in which each data point represents an individual placenta, and each color represents a different group (Index vs. Control) (Figure 1A). The closer the data points are to each other, the more closely related the transcriptional responses are. Moreover, to gain insight into similarities among replicates, the set of genes tested by qPCR was subjected to a hierarchical clustering procedure and presented as heatmaps (Figure 1B). The heatmap of the selected differential gene set shows the association of the biological samples into the two distinct groups (Index vs. Control). The heat map reveals that on average, mRNA expression within placental tissues of mothers clinically diagnosed as stressed (Index) was generally higher than the control group.

The qPCR analyses revealed differential gene expression between groups. Women with perceived symptoms of anxiety and depression during pregnancy had altered expression patterns for *CRH, HSD11B2,* and *AVP* genes in the placentas, compared to the control population. An upregulation of CRH (*p* < 0.05), *HSD11B2* (*p* < 0.05), and *AVP* (*p* < 0.001) gene expression was found in the Index group compared to the control, while *OGT* and *NR3C1* gene expression appeared similar between groups (Figure 2).

Additionally, Spearman correlation analyses were performed to investigate the association among the expression of all genes tested in this study (Table 3). Interestingly, a significant positive correlation between *CRH* with both *AVP* (Rho = 0.79; *p* < 0.001) and *HSD11B2* (Rho = 0.45; *p* < 0.03), and also between *AVP* with both *HSD11B2* (Rho = 0.6; *p* < 0.005) and NR3C1 (Rho = 0.56; *p* < 0.03) gene expression in the Control group was observed. Such correlations were not evident in the Index group, suggesting a possible loss of interaction in the mechanisms of action of these genes under stress circumstances.

### 3.2. Prenatal Stress Influence on Stress-Like Placental Protein Expression

The expression of proteins related to the significantly altered genes found in this study was analyzed by Western blotting. CRH and AVP proteins were clearly detected (Figure 3A–D) with bands of 22 and 17 kDa found in placental tissue of either Index or Control groups corresponding to CRH (anti-CRH; rabbit polyclonal antibody; LSBio-B11889) and AVP (anti-AVP polyclonal antibody; MBS9205129) proteins, respectively (Figure 3A,C,D). Significant changes in the expression of these proteins was not observed between groups. However, a trend for higher CRH protein levels in female placental samples compared to males was found following the line of CRH gene expression analyses (Figure 3B).

### 3.3. Prenatal Stress Influences the Association of Placental HSD11B2 Gene Expression with Hair Cortisol Levels

A significant positive correlation between *HSD11B2* (Rho = 0.54; *p* < 0.001) gene expression in term-placentas and maternal HCC-levels at parturition was found in the Index group (Table 4). This finding indicates a clear positive feedback between cortisol levels during labor and this cortisol-modulatory enzyme.

### 3.4. Placental Sex Depicts Differences in the Gene Expression of HSD11B2

The impact of offspring sex on stress-like gene expression among placental samples harvested from Index and Control women at term was further evaluated. These analyses demonstrated different responses on gene expression levels between male and female placentas (Figure 4A–E). However, only HSD11B2 showed greater (*p* < 0.05) levels of expression in males than females in the Index group (Figure 4C).

### 3.5. Offspring Birth Weight (BW) and Placental Gene Expression

In the present study the offspring BW was not affected by prenatal stress. Additionally, BW did not significantly correlate with placental gene expression in any of the groups examined (Table 5).

## 4. Discussion

The current study reports the effects of maternal stress during pregnancy on gene and protein expression levels of stress-related molecules in term-placentas. Amongst the most interesting results from this study, we found that the *CRH* gene doubled its expression in placenta samples from Index women compared to the Control group. It is known that, during pregnancy, CRH is responsible for preparing the environment for childbirth, thus influencing developmental trajectories towards this event. However, less is known about the gene encoding this stress-related hormone. Although it has been reported as expressed in human placenta under physiological circumstances, this is, to the best of our knowledge, the first study to report an overexpression of this gene in term-placentas under maternal stress influence. CRH is considered the central upstream mediator of stress pathway activation [34,35], and has been associated with concentration-dependent effects upon the immune system [36]. The elevation of CRH secretion in the presence of psychological stressors cause, well-before a recorded increase in glucocorticoid downstream, the release of cytokines and its associated fever [37]. Prenatal maternal stress during the first trimester is accompanied not only by CRH increases but also elevations of the pro-inflammatory cytokines IL-6 and TNFα in blood, suggesting a linkage between stress and immune activation that could potentially affect fetal programming [38,39]. In contrast, a body of evidence support the concept that CRH is capable of downregulating the immune system by decreasing T cell proliferation and natural killer (NK) cell cytotoxicity [40]. Peripherally, CRH can also act as an anti-inflammatory molecule reducing inflammatory exudate volume in various disease models [41]. Despite these apparent immunostimulant or immunosuppressive actions, the effects of CRH on the immune system are complex, and time- and tissue-specific [42,43]. While it is generally accepted that glucocorticoids suppress immune responses in the acute phase on any inflammatory process, their lengthy presence before an immune challenge can issue an inverse effect, depending on the tissue in question. For example, while chronic elevations of glucocorticoids suppress the peripheral immune system, they can promote a pro-inflammatory state on the immune cells in the brain [44]. Our findings support the notion that gene levels of placental *CRH* increase under prenatal stress conditions and that might have an impact on subsequent immune system process in the child, thus affecting further systematic development. However, follow-up studies of relevant immune-related genes in term-placentas are required.

Moreover, the action of CRH can be potentiated by vasopressin, oxytocin, epinephrine, norepinephrine, and angiotensin II as previously reported [45,46]. We studied the Arginine-Vasopressin stress-hormone (*AVP*) gene expression and its association with *CRH* levels in at term-placental samples exposed to prenatal stress. The AVP system rules the homeostasis of vascular tonus, fluid balance, as well as the endocrine stress responses [47] through several pathways. Firstly, AVP regulates water absorption via the posterior pituitary. Secondly, AVP is critically involved in the hypothalamic–pituitary–adrenal (HPA) stress axis via the posterior pituitary and thirdly AVP, by remaining in the central nervous system, contributes to behavior and cognitive functions [48]. Furthermore, AVP is important in the control of fetoplacental blood pressure and facilitates the transition of the newborn to air breathing, cardiovascular adaptation, thermogenesis, glucose, and water homeostasis [49]. As a response to acute hypoxia the human fetus actively secretes AVP to redistribute ventricular output towards the placenta [50]. Consistent with our results, AVP shows a peak of expression under fetal “stress” circumstances, such as heat stress, leading to widespread effects on fetal cardiovascular, renal, and lung functions [51]. The marker of AVP secretion Copeptin appears increased in pre-eclampsia, both in human and murine pregnancies [52]. In pregnant mice, infusion of AVP causes hypertension and renal glomerular endotheliosis, issuing placental oxidative stress which alters placental morphology, production of placental growth factor (PGF), and placental gene expression leading to intrauterine growth restriction, all mimicking dysfunctions seen during human pre-eclampsia [53]. In the present study, we found an upregulation of placental *AVP* expression in Index women compared to the Control group, suggesting that not only direct fetal stress but also maternal stress during pregnancy may be capable of inducing higher levels of this stress-like hormone probably leading to placental hypoxia and future adverse physiological functions. Furthermore, we observed a positive significant correlation in the expression of *CRH* and *AVP* genes in the Control group but not in the Index group. Thus, our data support the notion that there is a positive feedback between the expression of these two genes in the placenta under physiological pregnancies which is altered under maternal-induced stress. Additionally, we found a significant increment in the gene expression of the 11-β hydroxysteroid dehydrogenase (*HSD11B2*), that encodes 11ß-HSD2, the enzyme responsible for conversion of cortisol into inactive cortisone, in Index placentas compared to the Control group. Placental 11ß-HSD2 buffers the impact of maternal glucocorticoid exposure by converting cortisol/corticosterone into inactive metabolites [21,54], thus preventing the activation of glucocorticoid receptors [21,22]. However, previous studies indicate that maternal adversity including stress during pregnancy can lead to a dysfunction of this enzyme [55]. The methylation of the placental *HSD11B2* concurs with a dysfunctional neurobehavior among newborns born from mothers suffering from antenatal depression or anxiety [56]. In mice, mutation of the *HSD11B2* gene leads to hypertension, and increased anxiety-like behavior in adulthood [57], suggesting the regulation of placental 11ß–HSD2 function is central, linking antenatal stress with offspring morbidity long after birth. In the present study, we observed an upregulation of this gene as the levels of maternal cortisol increased during parturition in Index women, but not in the Control group, suggesting that higher levels of this enzyme are needed under stress condition in an attempt to block the transfer of cortisol within the fetal compartment.

Interestingly, *HSD11B2* gene expression was positively correlated with *CRH* and *AVP*, as well as *AVP* gene expression was also positively correlated to the glucocorticoid receptor (*NR3C1*), a nuclear receptor to which cortisol binds, in the Control group but not in the Index group, suggesting a clear dysregulation of stress-related gene interactions under prenatal stress circumstances.

Additionally, while several diseases associated with prenatal stress exhibit sex bias [58], we still lack information of how antenatal stress affects placental function of male or female sex fetuses. To further investigate potential mechanisms underlying the sex-specific effect of maternal stressors on placental function, we examined whether our target genes and proteins were differently expressed depending on the sex of the offspring. Interestingly, *HSD11B2* showed a significant increase in expression in male compared to female placentas in the Index group. Previous studies in rodents and humans reported that the activity and sensitivity of placental 11βHSD2 to maternal stimuli are sex-specific [59,60]. A wealth of data supports that stress-like disorders are sex-biased, being more common in women than in men [61,62,63]. The dysregulated state of hyperarousal, which disturbs sleep and leads to concentration problems and hyperactivity, and adds to symptoms of stress, anxiety, and depression [64], is more pronounced in females than males [65,66]. In particular, a major brain arousal center, the noradrenergic locus coeruleus (LC), appears to be more activated in females than males during emotion-evoking tasks, reinforcing the statement that a stressful event may elicit a greater LC-mediated arousal response in women than in men [67]. We here suggest that, besides the evidence that biological factors can increase female vulnerability to stress and stress-related pathologies [68,69], also, females are more sensitive to the transmission of maternal stress through the placenta under the regulation of certain stress-related genes. Particularly, in normal pregnancies, placental 11β-HSD2 activity is significantly higher in female than male fetuses [60]. However, we found a significant downregulation of the expression of this gene in female compared to male placentas in the Index group, which indicates that females might be subjected to a misfunction of the cortisol-blocking potential of this gene, thus being more exposed to a cortisol outbreak with potential developmental risks for the newborn. Although the mechanisms underlying this sex-specific pattern in maternal stress transmission are not clear, our results support the idea that males are more capable of circumventing the effects of maternal stressors by strengthening the maintenance of gene expression levels under physiological concentrations.

Overall, these findings provide novel evidence for the association of perceived maternal anxiety and depression during pregnancy and the dysregulation of placental gene expression of CRH, AVP, and HSD11B2 as potential mechanisms underlying adverse physiological and neurodevelopmental consequences for the newborn ultimately contributing to disease risk. Moreover, the sex-specificity of stress-related HSD11B2 gene regulation found in this study may provide new insights by which sex biases in neurodevelopmental programing occurs, leading to the identification of novel targets for therapeutic development.

## Figures and Tables

**Figure 1 genes-11-00869-f001:**
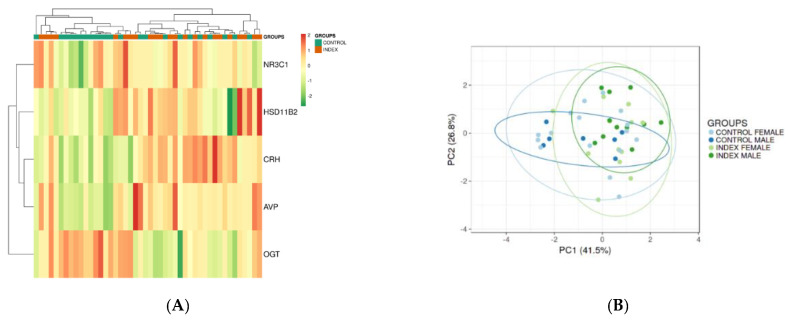
(**A**) Heat map plot of gene expression data of term-placenta samples of patients indicating symptoms of depression and anxiety (Index; *n* = 23) and patients with no symptoms of depression or anxiety (Control, *n* = 23). Both rows and columns are clustered using correlation distance and average linkage. Colors represent mRNA levels (red: higher, green: lower). (**B**) Principal component analysis (PCA) plot of gene expression data of male and female term-placenta samples of patients indicating symptoms of depression and anxiety (Index; *n* = 23) and patients with no symptoms of depression or anxiety (Control, *n* = 23). SVD (singular value decomposition) with imputation is used to calculate principal components. X and Y axis shows principal component 1 and principal component 2 that explain 41.5% and 26.8% of the total variance, respectively.

**Figure 2 genes-11-00869-f002:**
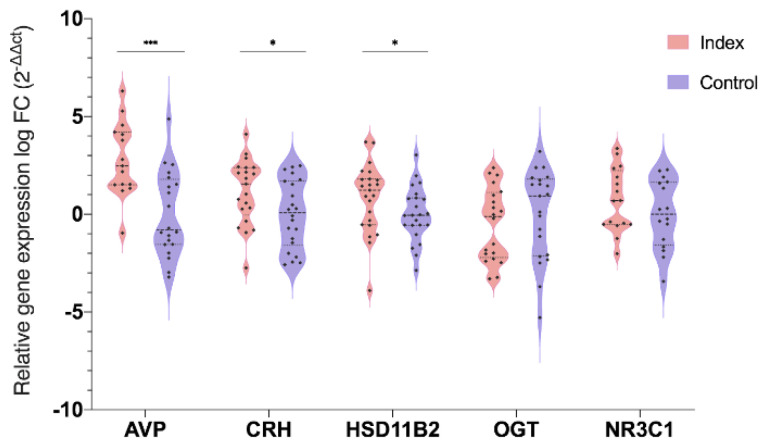
Differential gene expression of stress-related genes (Arginine Vasopressin—*AVP*; corticotropin-releasing hormone—*CRH*; 11β-hydroxysteroid dehydrogenase—*HSD11B2*; O-GlcNAc transferase—*OGT*; glucocorticoid receptor—*NR3C1*) in term-placenta samples of patients indicating symptoms of depression and anxiety (Index) and patients with no symptoms of depression or anxiety. The gene names are indicated on the X axis and the value on the Y axis represents the gene expression level in the binary logarithm (log 2) value. Data are presented by violin plot showing median and inter-quartile range (Q1–Q3). Asterisks indicate significant differences among groups (* *p* < 0.05; *** *p* < 0.001).

**Figure 3 genes-11-00869-f003:**
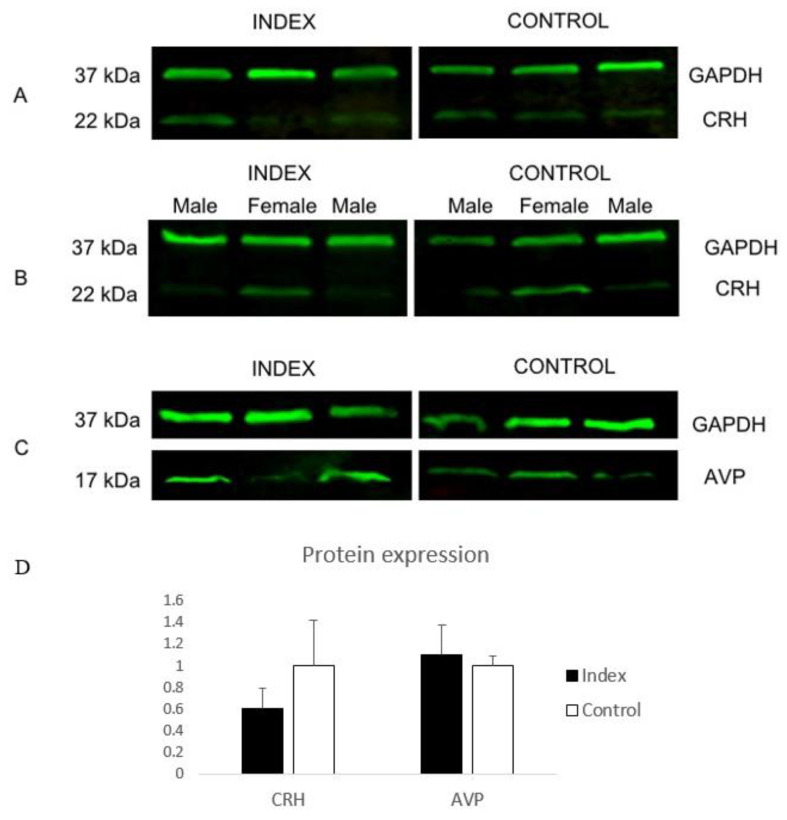
Western blot (WB) detection of the corticotropin-releasing hormone (CRH) and Arginine Vasopressin (AVP) proteins in term-placentas of patients indicating symptoms of depression and anxiety (Index) and patients with no symptoms of depression or anxiety (Control). (**A**) Human anti-CRH polyclonal antibody (LS-B11889) tested in Index and Control group and (**B**) between males and females identified expected bands at 22 kDa. In (**C**) the human anti-AVP polyclonal antibody (MBS9205129) tested in Index and Control groups identified expected bands at 17 kDA. (**D**) Graphical representation of protein expression of *CRH* and *AVP* between Index and Control groups.

**Figure 4 genes-11-00869-f004:**
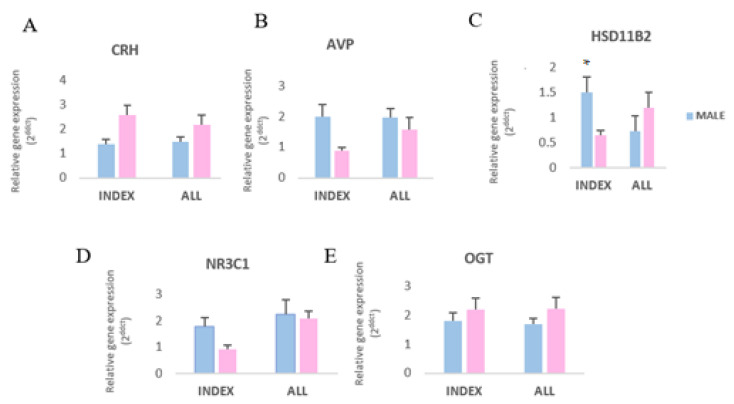
Differential gene expression of stress-related genes (**A**: corticotropin-releasing hormone—*CRH*; **B**: Arginine Vasopressin—*AVP*; **C**: 11β-hydroxysteroid dehydrogenase—*HSD11B2*; **D**: glucocorticoid receptor—*NR3C1*; **E**: O-GlcNAc transferase—*OGT*) in male and female term-placentas of patients indicating symptoms of depression and anxiety (Index) and all patients examined. Asterisks indicate significant differences among groups (* *p* < 0.05).

**Table 1 genes-11-00869-t001:** Demographic data of patients indicating symptoms of depression and anxiety (Index) and patients with no symptoms of depression nor anxiety (Control).

	Index	Controls	
	Mean/SD	Mean/SD	*p*-Value
**Maternal age**	29.5/4.9	28.5/4.2	0.483
**BMI**	26.4/4.7	26.6/5.3	0.891
**Gestational age**	39.6/1.2	40.0/1.1	0.312
	***n* (%)**	***n* (%)**	
**Ethnicity**			0.665
Swedish	21 (91.3)	19 (82.6)	
Non-Swedish	2 (8.7)	4 (17.4)	
**Parity**			**NA**
Primi	8 (34.8)	8 (34.8)	
Multi	15 (65.2)	15 (65.2)	
**Delivery mode**			1.000
PN	20 (87.0)	21 (91.3)	
CS or Instrumental	3 (13.0)	2 (8.7)	
**Gender of the child**			0.227
Girl	16 (69.6)	12 (52.2)	
Boy	7 (30.4)	11 (47.8)	

**Table 2 genes-11-00869-t002:** Sequence of primers used for real-time quantitative PCR.

Gene Name	Primers (5′-3′)	Accession No	Amplicon Size (bp)
*CRH*	F:GAGAGAGGGAGAGAGCCTATAC R:TGACCAAGGACTGGAAAGATG	NC_00008.11	320
*OGT*	F:GGCTGACCAGTTAGAGAAGAATAG R:TGCCTGGAATAGACTGCATAAG	NC_000023.11	260
*HSD11B2*	F:TGCTTCAAGACAGAGTCAGTG R:GGCATCTACAACTGGGGTGA	NC_000016.10	183
*GAPDH*	F:GGAAGGTGAAGGTCGGAGTC R:GAGGGATCTCGCTCCTGGAA	NC_000012.11	244
*AVP*	Bio-Rad ID: qHsaCED0021009	NC_000020.10	78
*NR3C1*	Bio-Rad ID: qHsaCEP0050768	NT_029289.11	118

**Table 3 genes-11-00869-t003:** Spearman correlations between gene expression data of stress-related genes (corticotropin-releasing hormone—*CRH*; Arginine Vasopressin—*AVP*; 11β-hydroxysteroid dehydrogenase—*HSD11B2*; glucocorticoid receptor—*NR3C1*; O-GlcNAc transferase—*OGT*) in term-placentas of patients indicating symptoms of depression and anxiety (Index), patients with no symptoms of depression nor anxiety (Control), and all patients examined in this study (All). Comparisons showing statistical significance are marked in bold font.

	*AVP*			*HSD11B2*			*OGT*			*NR3C1*		
	Index	Control	All	Index	Control	All	Index	Control	All	Index	Control	All
*CRH*	Rho = 0.29	**Rho = 0.79**	**Rho = 0.6**	Rho = −0.23	**Rho = 0.45**	Rho = 0.2	Rho = −0.36	Rho = −0.38	**Rho = −0.46**	Rho = −0.09	Rho = 0.23	Rho = 0.12
	*p* = 0.3	***p* < 0.001**	***p* < 0.001**	*p* = 0.31	***p* = 0.03**	*p* = 0.22	*p* = 0.12	*p* = 0.09	***p* = 0.003**	*p* = 0.75	*p* = 0.37	*p* = 0.52
	*n* = 15	***n* = 20**	***n* = 35**	*n* = 21	***n* = 22**	*n* = 43	*n* = 19	*n* = 20	***n* = 39**	*n* = 14	*n* = 16	*n* = 30
*AVP*				Rho = 0.08	**Rho = 0.6**	**Rho = 0.56**	Rho = −0.16	Rho = −0.41	Rho = −0.33	Rho = 0.007	**Rho = 0.56**	Rho = 0.36
				*p* = 0.76	***p* = 0.005**	***p* < 0.001**	*p* = 0.57	*p* = 0.09	*p* = 0.06	*p* = 0.98	***p* = 0.03**	*p* = 0.07
				*n* = 15	***n* = 20**	***n* = 35**	*n* = 15	*n* = 18	*n* = 33	*n* = 12	***n* = 15**	*n* = 27
*HSD11B2*							Rho = 0.29	Rho = 0.09	Rho = 0.11	Rho = 0.12	Rho = 0.34	Rho = 0.28
							*p* = 0.21	*p* = 0.69	*p* = 0.5	*p* = 0.64	*p* = 0.2	*p* = 0.12
							*n* = 20	*n* = 21	*n* = 41	*n* = 16	*n* = 16	*n* = 32
*OGT*										Rho = 0.09	Rho = 0.009	Rho = 0.008
										*p* = 0.73	*p* = 0.97	*p* = 0.96
										*n* = 15	*n* = 16	*n* = 31

**Table 4 genes-11-00869-t004:** Spearman correlations between maternal hair cortisol levels (at week 24–25 of pregnancy, parturition and 8 weeks postpartum), and gene expression data of stress-related genes (corticotropin-releasing hormone—*CRH*; Arginine Vasopressin—*AVP*; 11β-hydroxysteroid dehydrogenase—*HSD11B2*; glucocorticoid receptor—*NR3C1*; O-GlcNAc transferase—*OGT*) in term-placentas of patients indicating symptoms of depression and anxiety (Index), patients with no symptoms of depression nor anxiety (Control) and all patients examined in this study (All). Comparisons showing statistical significance are marked in bold font.

GROUP	CORTISOL MEASUREMENTS	CRH	*OGT*	*HSD11B2*	*AVP*	*NR3C1*
**Index**	Week 24–25 of pregnancy	Rho = 0.37	Rho = 0.2	Rho = 0.021	Rho = −0.35	Rho = 0.31
		*p* = 0.09	*p* = 0.38	*p* = 0.92	*p* = 0.19	*p* = 0.24
		*n* =21	*n* = 20	*n* = 23	*n* = 15	*n* = 16
	Parturition	Rho = 0.34	Rho = 0.02	**Rho = 0.54**	Rho = 0.37	Rho = 0.29
		*p* = 0.12	*p* = 0.99	***p* = 0.007**	*p* = 0.17	*p* = 0.27
		*n* = 21	*n*=20	***n* = 23**	*n* = 15	*n* = 16
	8 weeks Postparturition	Rho = 0.45	Rho = 0.09	Rho = 0.29	Rho = −0.12	Rho = −0.24
		*p* = 0.06	*p* = 0.72	*p* = 0.22	*p* = 0.68	*p* = 0.27
		*n* = 17	*n* = 17	*n* = 19	*n* = 13	*n* = 15
**Control**	Week 24–25 of pregnancy	Rho = −0.04	Rho = −0.19	Rho = −0.17	Rho = 0.21	Rho = 0.171
		*p* = 0.84	*p* = 0.4	*p* = 0.44	*p* = 0.36	*p* = 0.52
		*n*=22	*n* = 21	*n* = 23	*n* = 20	*n* = 16
	Parturition	Rho = 0.03	Rho = −0.08	Rho = 0.03	Rho = −0.017	Rho = 0.46
		*p* = 0.88	*p* = 0.7	*p* = 0.87	*p* = 0.94	*p* = 0.06
		*n* = 22	*n* = 20	*n* = 22	*n* = 20	*n* = 16
	8 weeks Postparturition	Rho = −0.14	Rho = −0.32	Rho = −0.23	Rho = −0.27	Rho = −0.15
		*p* = 0.5	*p* = 0.18	*p* = 0.32	*p* = 0.28	*p* = 0.6
		*n* = 19	*n* = 19	*n* = 20	*n* = 17	*n* = 14
**All**	Week 24–25 of pregnancy	Rho = 0.15	Rho =- 0.05	Rho = −0.006	Rho = 0.019	Rho = 0.24
		*p* = 0.33	*p* = 0.75	*p* = 0.96	*p* = 0.91	*p* = 0.19
		*n* = 43	*n* = 41	*n* = 46	*n* = 35	*n* = 32
	Parturition	Rho = 0.17	Rho = −0.09	Rho = 0.28	Rho = 0.044	**Rho = 0.4**
		*p* = 0.25	*p* = 0.57	*p* = 0.054	*p* = 0.8	***p* = 0.023**
		*n* = 43	*n* = 40	*n* = 45	*n* = 35	***n* = 32**
	8 weeks Postparturition	Rho = 0.04	Rho = −0.065	Rho = 0.01	Rho = −0.35	Rho = −0.16
		*p* = 0.8	*p* = 0.71	*p* = 0.94	*p* = 0.052	*p* = 0.4
		*n* = 36	*n* = 36	*n* = 39	*n* = 30	*n* = 29
	Parturition	Rho = 0.28	Rho = −0.05	Rho = −0.11	Rho = 0.08	Rho = 0.26
		*p* = 0.07	*p* = 0.74	*p* = 0.45	*p* = 0.65	*p* = 0.16
		*n* = 42	*n* = 40	*n* = 45	*n* = 34	*n* = 31
	8 weeks Postparturition	Rho = 0.23	Rho = −0.12	Rho = 0.07	Rho = −0.14	Rho = 0.17
		*p* = 0.19	*p* = 0.49	*p* = 0.66	*p* = 0.47	*p* = 0.38
		*n* = 31	*n* = 32	*n* = 34	*n* = 27	*n* = 26

**Table 5 genes-11-00869-t005:** Spearman correlations between offspring birth weight (BW) and gene expression data of stress-related genes (corticotropin-releasing hormone—*CRH*; Arginine Vasopressin—*AVP*; 11β-hydroxysteroid dehydrogenase—*HSD11B2*; glucocorticoid receptor—*NR3C1*; O-GlcNAc transferase—*OGT*) in term-placentas of patients indicating symptoms of depression and anxiety (Index), patients with no symptoms of depression nor anxiety (Control), and all patients examined in this study (All).

BW	PLACENTAL GENE EXPRESSION LEVELS AT TERM				
	*CRH*	*OGT*	*HSD11B2*	*AVP*	*NR3C1*
**Index**	Rho = −0.25	Rho = −0.1	Rho = −0.021	Rho = −0.25	Rho = −0.03
	*p* = 0.28	*p* = 0.67	*p* = 0.93	*p* = 0.37	*p* = 0.92
	*n* = 21	*n* = 20	*n* = 23	*n* = 15	*n* = 16
**Control**	Rho = −0.04	Rho = 0.25	Rho = 0.2	Rho = 0.14	Rho = −0.03
	*p* = 0.87	*p* = 0.91	*p* = 0.36	*p* = 0.57	*p* = 0.91
	*n* = 22	*n* = 21	*n* = 23	*n* = 20	*n* = 16
**All**	Rho = −0.18	Rho = 0.01	Rho = 0.042	Rho = −0.11	Rho = −0.03
	*p* = 0.26	*p* = 0.96	*p* = 0.78	*p* = 0.54	*p* = 0.88
	*n* = 43	*n* = 41	*n* = 46	*n* = 35	*n* = 32

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
