# Peer review of "Expression of Stress-Mediating Genes is Increased in Term Placentas of Women with Chronic Self-Perceived Anxiety and Depression"

_genes, 2020, doi:10.3390/genes11080869_

Round 1
Reviewer 1 Report
The paper analysed placenta gene and protein expression and hair cortisol levels in women that apparently were suffering from anxiety and depression. The study provides some novel results, but their significance needs to be properly discussed (see comment 8) and some technical issues need to be addressed:
1) Lines 85-86: In the final paragraph of the introduction the authors indicated that they “determined the term-placental expression of main glucocorticoid pathway genes…..as well as other key stress biomarkers…. and their association with maternal and fetal HCC levels, offspring gender and birth weight”. However, the authors did not present data on fetal HCC levels and birth weight. This need to be added, especially the birth weight data. The latter will truly validate their model, this is, assuming their found altered birth weights (low birth weights) in women with anxiety and depression.
2) The authors need to provide information on the primers used in qPCR (i.e. provide a table with a list of primers used).
3) The authors also need to provide details regarding the quantification of protein expression. In the current format it gives the impression that their protein expression results were obtained just by looking at the blots. They need to present graphs with arbitrary units analysed (i.e. statistical analysis).
4) Please use the same format used in figure 2 for figure 4 (i.e. violin plots). And provide figures with better resolution (they are blurry in their current format).
5) Line 274: “Figure 3A” should be “Figure 3” (remove the A).
6) Lines 282-285: This sentence does not have a proper syntax, please rephrase it.
7) Lines 290-298: This paragraph needs to be rephrased to indicate clearly that the only statistical difference was with HSD11B2 in index patients. Please do not indicate there was “trend” unless you provide some statistical support (e.g. P< 0.09). But my advice will be to avoid making this type of statements (i.e. avoid to highlight numerical differences that do have statistical significance).
8) The authors need to tone down the interpretation of their results. Some of the statements made in the discussion are not back up by their results. For instance, in lines 388-389 the authors stated that CRH and OGT gene expression was higher in placentas from female offspring, but the results in figure 4 did not indicate any statistical difference in this genes (please see comment 7). Also, they put a major emphasis in explaining the differences in gene expression for CRH and AVP and the possible implications of this altered gene expression, but their protein expression analysis did not show any differences for these two genes (i.e. index vs control in figure 3, pictures A and C). It is known that gene expression does not always correlate with protein expression and hence the usefulness of gene expression should be put into perspective. By ignoring the protein expression results the authors are overinterpreting gene expression results. The authors need to address this in their discussion and be cautious on their conclusions.
Author Response
Lines 85-86: In the final paragraph of the introduction the authors indicated that they “determined the term-placental expression of main glucocorticoid pathway genes…..as well as other key stress biomarkers…. and their association with maternal and fetal HCC levels, offspring gender and birth weight”. However, the authors did not present data on fetal HCC levels and birth weight. This need to be added, especially the birth weight data. The latter will truly validate their model, this is, assuming their found altered birth weights (low birth weights) in women with anxiety and depression.
Reply: We thank the reviewer for her/his observation and suggestions. We have only studied the correlation between gene expression and maternal HCC but did not study fetal HCC. Therefore, we have now modified the paragraph accordingly. We have also added a new table with the data from the correlation analysis between gene expression and offspring birth weight (Table 5). We have also clarified in the results section that we did not find significant changes in birth weights between Index and Control group (Lines 363-364 of the revised Ms).
The authors need to provide information on the primers used in qPCR (i.e. provide a table with a list of primers used).
Reply: A table with the primer sequences (Table 2) has been added, and the matter has been specified in the materials and methods section (Line 176 of the revised Ms).
The authors also need to provide details regarding the quantification of protein expression. In the current format it gives the impression that their protein expression results were obtained just by looking at the blots. They need to present graphs with arbitrary units analysed (i.e. statistical analysis).
Reply: Numeric data of the signal of each band in the blot-images were obtained using the Image Studio 4.0 software (LI-COR Biosciences) after scanning the membranes. Then, data were normalized with its endogenous control andstatistical significance between Index group (n=3) and the Control group (n=3) was determined using Student's t-test. We have now included a graph in Figure 3 to complete the blot images and have clarified this information in the material and methods section (Lines 203-204 of the revised Ms).
Please use the same format used in figure 2 for figure 4 (i.e. violin plots). And provide figures with better resolution (they are blurry in their current format).
Reply: Violin plots (Fig 2) were best to represent the distribution of the individual presented hereby, against graphical bars (Fig 4) which give a view of differences in gene expression between offspring sex. The data in Fig 4 clearly mark the low number of male placentas in the control group (7 out of 23). As well, we have modified the figures aiming proper visualization of the data.
Line 274: “Figure 3A” should be “Figure 3” (remove the A).
Reply: Done (Line 314 of the revised Ms).
Lines 282-285: This sentence does not have a proper syntax, please rephrase it.
Reply: Done (Lines 323-326 of the revised Ms).
Lines 290-298: This paragraph needs to be rephrased to indicate clearly that the only statistical difference was with HSD11B2 in index patients. Please do not indicate there was “trend” unless you provide some statistical support (e.g. P< 0.09). But my advice will be to avoid making this type of statements (i.e. avoid to highlight numerical differences that do have statistical significance).
Reply: We have clarified the statement (Lines 330-334 of the revised Ms).
The authors need to tone down the interpretation of their results. Some of the statements made in the discussion are not back up by their results. For instance, in lines 388-389 the authors stated that CRH and OGT gene expression was higher in placentas from female offspring, but the results in figure 4 did not indicate any statistical difference in this genes (please see comment 7). Also, they put a major emphasis in explaining the differences in gene expression for CRH and AVP and the possible implications of this altered gene expression, but their protein expression analysis did not show any differences for these two genes (i.e. index vs control in figure 3, pictures A and C). It is known that gene expression does not always correlate with protein expression and hence the usefulness of gene expression should be put into perspective. By ignoring the protein expression results the authors are overinterpreting gene expression results. The authors need to address this in their discussion and be cautious on their conclusions.
Reply: We agree with the reviewer that focusing only on statistically significant differences among groups make the discussion more robust. In the revised version of the manuscript we have avoided statements about results showing only trends, including the statements made concerning CRH and OGT differences in female placentas. We also agree with the reviewer that gene expression does not always correlate with protein expression, and vice-versa, considering post-transcriptional protein structural changes and the involvement of multiple genes for some/most proteins. In any case, the expression of the genes on all samples used in this study (n=23 per group) was clearly altered. In the preliminary protein analysis using a restricted number of samples (n=3) we did not find a clear dysregulation of protein expression and so we did not include the rest of the material, but consider this data as relevant to present. Such data prompt us to consider further analysis on protein expression, studying not only maternal stress condition of the Index group but also other health conditions, demography, familiar antecedents and other parameters.
Reviewer 2 Report
Over the past decade, developmental origins of health and disease have attracted increasing attention. The placenta as the primary messenger system between mother and fetus plays a key role in mediating stress factors to the fetus. The capacity of the placenta to protect the fetus from environmental insults may also vary in a sex dependent manner. Placental gene expression is therefore key to the understanding of placental responses in an individual fetus‘ development. This manuscript entitled "Expression of stress-mediating genes is increased in term placentas of women with chronic self-perceived anxiety and depression” presents novel and interesting data elegantly linking anxiety and depression inventories, cortisol measurements as maternal stress indicators and placental expression analysis of selected stress genes.
Key findings include a dysregulation of gene expression for CRH, AVP and HSD11B2 (the stress-modulating enzyme 11β-hydroxysteroid dehydrogenase) genes in the group of patients with an elevated level of anxiety and maternal stress, while placental gene expression of (HSD11B2) was related to both hair cortisol levels and the sex of the newborn in pregnancies perceived as stressful.
The study has been adequately carried out. The methods and the data are presented clearly and in a precise manner, the data is discussed extensively. However, due to the descriptive character of this study and the limited number of genes investigated, a complete mechanistic insight into maternal stress-mediated placental gene expression patterns does not emerge from this study. Moreover, it would be interesting to know whether therapeutic interventions such as talk-therapy in pregnancies perceived as stressful would also alter placental gene expression.
Nevertheless, the study contributes to our understanding of the significance of maternal stress during pregnancy.
Author Response
We thank the reviewer for the kind comments. We agree with the reviewer that this study can be considered a preliminary study with some interesting data that needs to be further expanded with other experiments.
Reviewer 3 Report
In submitted manuscript authors investigated the association between placental expressions of certain glucocorticoid pathway genes as well as key stress markers, and chronical stress and depression during pregnancy. Overall, the subject is interesting however the placental expressions of two targeted genes, HSD11B2 and NR3C1, in depressive gestations were already reported (Zhang W. et. al. Infancy Mar-Apr 2018;23(2):211-231 and Capron LE. et. al. Psychoneuroendocrinology. 2018 Jan; 87:166-172). There are concerns regarding the study design, data presentation and interpretation of data. Here are major comments in this regard:
- It is unclear why the conservative EPDS score >10 was considered as depression threshold whereas at least 2 other independent studies on Swedish communities proposed scores at ≥11.5 (Acta Psychiatr Scand. 1996 Sep;94(3):181-4.) and more recently ≥13 (Nord J Psychiatry. 2011 Dec;65(6):414-8.), as the optimal cut-offs. Since the optimal cutoff scores of the EPDS varies considerably among populations, authors should reason or provide a valid reference for their criteria in studied community. The provided citation (Cox et al., 1987) does not exist in “References”.
- Similarly, Beck Anxiety Inventory (BAI) score of >10 was considered as indicator of anxiety in this study however the BAI score of 8-15 is defined for mild anxiety (Beck AT, Steer RA. BAI Beck Anxiety Inventory Manual Swedish Version. Stockholm: Psykologiförlaget; 2005.). Psychological disorders are of spectrum nature so more stringent selection of subjects from extreme sides of spectrum, could help to obtain results with better contrasts and draw stronger conclusions, especially in biological/molecular studies. The selection strategy in current study would explain the big overlap of control/index data point on PCA plot in Fig. 1-A.
- Authors have to clearly state the pregnancy inclusion/exclusion criteria. Were pregnancy complications (like preeclampsia) and/or preterm birth, considered in the population selection? What about metabolic disorders like gestational diabetes?
- Providing a demographic table providing necessary data such as Maternal age, ethnicity (if applicable), BMI, Gravidity, Parity, gestational age at delivery, delivery mode, fetal gender etc. for both index and controls, would help reader to better understand the studied population.
- Since the data clearly show the significant effect of fetal sex on gene expression patterns (Fig. 4), it will be informative to add an extra strip (above/below GROUPS) in Fig. 1-A that presents the fetal sex.
- Why unequal numbers (two males and one female) were presented in Fig. 3? Is this the only WB experiment? Or a representative gel? How many samples were examined by WB? Were any semi-quantitative data generated for statistical analysis and a bar graph?
- 4 is confusing. What does the “all patients examined” mean?
- It seems the important role of fetal sex in current study was not investigated as deserved. According to Fig. 4, 4 out of 5 studied genes were differentially expressed in males compared to females, only in index group however nothing was discussed in this regard.
- The discussion is poorly structured and includes over-interpretation of provided data, in occasions.
- In the first one third of section, authors discussed the association between CRH and immune system which is out of scope for current study.
- Line 374-376: “These studies suggest that the regulation of placental 11ß–HSD2 levels may be a mechanistic link between the experience of maternal gestational stress and long-term health outcomes in offspring.”. How did authors show the “long-term health outcomes in offspring”?
Author Response
It is unclear why the conservative EPDS score >10 was considered as depression threshold whereas at least 2 other independent studies on Swedish communities proposed scores at ≥11.5 (Acta Psychiatr Scand. 1996 Sep;94(3):181-4.) and more recently ≥13 (Nord J Psychiatry. 2011 Dec;65(6):414-8.), as the optimal cut-offs. Since the optimal cutoff scores of the EPDS varies considerably among populations, authors should reason or provide a valid reference for their criteria in studied community. The provided citation (Cox et al., 1987) does not exist in “References”.
Reply: We apologize for failing in including Cox et al 1987 (Cox JL, Holden JM, Sagovsky R. Detection of postnatal depression: Development of the Edinburgh Postnatal Depression Scale. Br J Psychiatry 1987; 150: 782–6) in the original Reference list, particularly since the authors proposed a cut-off level of 10 for screening purposes in the post-natal period. As we have screened both during pregnancy and post-partum we chose to use the same cut-off. We have also used > 10 in earlier published screening studies (Josefsson A, Berg G, Nordin C, Sydsjö G. Prevalence of depressive symptoms in late pregnancy and postpartum. Acta Obstet Gynecol Scand 2001; 80: 251-255). By selecting the above threshold the sensitivity for the detection of major depression was nearly 100% and the specificity 82% (Harris B, Huckle P, Thomas R, Johns S, Fung H. The use of rating scales to identify postnatal depression. Br J Psychiatry 1989; 154: 813–17). This information, as well as the references have been included in the revised version of the manuscript.
Similarly, Beck Anxiety Inventory (BAI) score of >10 was considered as indicator of anxiety in this study however the BAI score of 8-15 is defined for mild anxiety (Beck AT, Steer RA. BAI Beck Anxiety Inventory Manual Swedish Version. Stockholm: Psykologiförlaget; 2005.). Psychological disorders are of spectrum nature so more stringent selection of subjects from extreme sides of spectrum, could help to obtain results with better contrasts and draw stronger conclusions, especially in biological/molecular studies. The selection strategy in current study would explain the big overlap of control/index data point on PCA plot in Fig. 1-A.
Reply: We chose to use the cut-off 10 in order to detect symptoms of anxiety, not for diagnostic purposes. The BAI contains 21 questions, each answer being scored on a scale value of 0 (not at all) to 3 (severely). Higher total scores indicate more severe anxiety symptoms. The standardized cut-offs are:
0–9: normal or no anxiety
10-18: mild to moderate anxiety
19-29: moderate to severe anxiety
30-63: severe anxiety.
Julian, Laura J. (7 November 2011). "Measures of Anxiety". Arthritis Care & Research. 63: S467–72. doi:10.1002/acr.20561. ISSN 2151-464X. PMC 3879951. PMID 22588767.
Authors have to clearly state the pregnancy inclusion/exclusion criteria. Were pregnancy complications (like preeclampsia) and/or preterm birth, considered in the population selection? What about metabolic disorders like gestational diabetes?
Reply: Women displaying pregnancy complications, including gestational diabetes, preeclampsia and/or preterm birth were excluded. The placenta results have been presented considering the sex of the newborns. Premature births are not included since premature birth was an exclusion criterium and they were assisted in another clinic, which prevented mistakes when collecting samples. The matter is presented in Table 1, and exclusion criteria are now clearly stated in the revised Ms version.
Providing a demographic table providing necessary data such as Maternal age, ethnicity (if applicable), BMI, Gravidity, Parity, gestational age at delivery, delivery mode, fetal gender etc. for both index and controls, would help reader to better understand the studied population.
Reply: We thank the reviewer for her/his suggestion. We have included a table with demographic information of the patients (Table 1).
Since the data clearly show the significant effect of fetal sex on gene expression patterns (Fig. 4), it will be informative to add an extra strip (above/below GROUPS) in Fig. 1-A that presents the fetal sex.
Reply: We thank the reviewer for the interesting suggestion. We have adapted Figure 1A following the recommendations.
Why unequal numbers (two males and one female) were presented in Fig. 3? Is this the only WB experiment? Or a representative gel? How many samples were examined by WB? Were any semi-quantitative data generated for statistical analysis and a bar graph?
Reply: We found a high overrepresentation of protein expression in some female placentas compared to male samples (as shown in the representative gel). However, this pattern was not stable among samples and was not significantly altered. In future experiments we will examine if there could be any other characteristic affecting protein expression in some females but not in others. WB analyses were performed with 3 samples per group.
4 is confusing. What does the “all patients examined” mean?
Reply: The group “ALL” includes both the Index and the Control group separated by male or female samples. We combined Index and Control groups to explore if the differences in gene expression between males and females were related to maternal stress or just to the sex of the newborn in itself.
It seems the important role of fetal sex in current study was not investigated as deserved. According to Fig. 4, 4 out of 5 studied genes were differentially expressed in males compared to females, only in index group however nothing was discussed in this regard.
Reply: Although the bar graphs show different patterns for male and female placentas in most of the genes analyzed, only HSD11B2 gene expression was significantly altered. To be sure we make the right conclusions out of our results, we only discussed the alterations on the genes clearly dysregulated.
The discussion is poorly structured and includes over-interpretation of provided data, in occasions.
In the first one third of section, authors discussed the association between CRH and immune system which is out of scope for current study.
Reply: We have now modified some statements made in the discussion in order to be less speculative. However, we think that the possible impact of CRH gene dysregulation on the immune system of the offspring is an important fact to be mentioned in the present revised manuscript, considering the presence of dysfunctions with compromised immune capacity in adult life. We aim to follow up this line of research looking at immuno-related genes at term-placenta and during placentation.
Line 374-376: “These studies suggest that the regulation of placental 11ß–HSD2 levels may be a mechanistic link between the experience of maternal gestational stress and long-term health outcomes in offspring.”. How did authors show the “long-term health outcomes in offspring”?
Reply: We must clarify that we were referring to studies mentioned previously in the discussion, where the authors found neurobehavioral disfunctions during the adulthood, after maternal mutations of placental HSD11B2. We did not monitor health outcomes of the offspring in the present study. We have rewritten the text for a clearer interpretation of our conclusions.
Round 2
Reviewer 3 Report
Authors addressed all concerns. I have no more comment.